# Depressive episode and treatment outcomes in elderly individuals with tuberculosis: A prospective cohort study in Korea

Ahran Kim[1], Hyung Woo Kim[2], Ju Sang Kim[2], Jin Woo Kim[3], Yong Hyun Kim[4], Heayon Lee[5], Yeonhee Park[6], Jee Youn Oh[7], Hyeon-Kyoung Koo[8], Ock-Hwa Kim[9], Yun-Jeong Jeong[10], Yong-Soo Kwon[11], Won Yeon Lee[12], Yoolwon Jeong[13], Jinsoo Min [14]*

1 Division of Pulmonary and Critical Care Medicine, Department of Internal Medicine, St. Vincent's Hospital, College of Medicine, The Catholic University of Korea, Seoul, Republic of Korea, 2 Division of Pulmonary and Critical Care Medicine, Department of Internal Medicine, Incheon St. Mary's Hospital, College of Medicine, The Catholic University of Korea, Seoul, Republic of Korea, 3 Division of Pulmonary and Critical Care Medicine, Department of Internal Medicine, Uijeongbu St. Mary's Hospital, College of Medicine, The Catholic University of Korea, Seoul, Republic of Korea, 4 Division of Pulmonary and Critical Care Medicine, Department of Internal Medicine, Bucheon St. Mary's Hospital, College of Medicine, The Catholic University of Korea, Seoul, Republic of Korea, 5 Division of Pulmonary, Critical Care and Sleep Medicine, Department of Internal Medicine, Eunpyeong St. Mary's Hospital, College of Medicine, The Catholic University of Korea, Seoul, Republic of Korea, 6 Division of Pulmonary and Critical Care Medicine, Department of Internal Medicine, Daejeon St. Mary's Hospital, College of Medicine, The Catholic University of Korea, Seoul, Republic of Korea, 7 Division of Pulmonary, Allergy, and Critical Care Medicine, Department of Internal Medicine, Korea University Guro Hospital, Korea University College of Medicine, Seoul, Republic of Korea, 8 Division of Pulmonary and Critical Care Medicine, Department of Internal Medicine, Ilsan Paik Hospital, Inje University College of Medicine, Goyang, Republic of Korea, 9 Division of Pulmonology and Allergy, Department of Internal Medicine, Chungnam National University, Chungnam National University Sejong Hospital, Sejong, Republic of Korea, 10 Division of Pulmonary and Critical Care Medicine, Department of Internal Medicine, Dongguk University Ilsan Hospital, Goyang, Republic of Korea, 11 Division of Pulmonary and Critical Care Medicine, Department of Internal Medicine, Chonnam National University Hospital, Chonnam National University Medical School, Gwangju, Republic of Korea, 12 Division of Pulmonary and Critical Care Medicine, Department of Internal Medicine, Wonju Christian Hospital, Yonsei University Wonju College of Medicine, Wonju, Republic of Korea, 13 Department of Preventive Medicine, Dankook University Hospital, Cheonan, Republic of Korea, 14 Division of Pulmonary and Critical Care Medicine, Department of Internal Medicine, Seoul St. Mary's Hospital, College of Medicine, The Catholic University of Korea, Seoul, Republic of Korea

* minjinsoo@catholic.ac.kr

## Abstract

### Background

Depression is common in the elderly and has been linked with poor tuberculosis treatment outcomes.

### Methods

We conducted a prospective multicenter cohort study of adult aged ≥ 65 years with active tuberculosis in the Republic of Korea between 2020 and 2022. Sociodemographic and clinical data were obtained by interview. Depression was assessed using

**Data availability statement:** The ownership of the primary datasets lies with the Korea National Institute of Health (KNIH), Korea Disease Control and Prevention Agency (KDCA). The de-identified datasets generated and/or analyzed during the current study can be made available for replication purposes upon reasonable request. Interested researchers should contact Dr. Tae-hyoun Kim (white-vet81@korea.kr) at the Division of Bacterial Disease Research, Center for Infectious Disease Research, National Institute of Health, Korea Disease Control and Prevention Agency, Cheongju, the Republic of Korea. Requests must include a brief study proposal, institutional review board approval, and a data use agreement. Data access will be granted only with permission from the KNIH and will be limited to the scope of the approved research. The datasets originate from the KNIH.

**Funding:** This study work was supported by the Research Program funded by the Korea National Institute of Health (2020ER520502 & 2025E200100). The funder had no role in the study design, data collection and analysis, decision to publish, or preparation of the manuscript.

**Competing interests:** The authors have declared that no competing interests exist.

the Patient Health Questionnaire-9 (PHQ-9), with a score ≥10 indicating a depressive episode. Logistic regression analyses were conducted to identify factors associated with depressive episodes and evaluate their association with treatment outcomes.

## Results

Among the 361 elderly individuals with active tuberculosis who completed the questionnaire, 69 (19.1%) were classified as having a depressive episode. Depressive episodes were significantly associated with unemployment, higher comorbidity burden, and the presence of tuberculosis-related symptoms such as cough and constitutional symptoms. Functional impairment was reported in 39.6% (143/361) of all participants and increased with the greater severity of depressive symptoms. Suicidal ideation was observed in 19.7% (71/361) of all participants and was independently associated with alarming tuberculosis symptoms. Among the participants with rifampin-susceptible tuberculosis, those with depressive episodes had significantly lower treatment success rates (64.7% vs. 79.1%, p = 0.012). In multivariable logistic regression analysis, depression remained independently associated with reduced odds of treatment outcomes (adjusted odds ratio, 0.478; 95% confidence interval, 0.261–0.878).

## Conclusions

In elderly individuals with tuberculosis, depressive episodes are associated with functional impairment, suicidal ideation, and poor treatment outcomes. Routine mental health assessments at tuberculosis diagnosis may help improve clinical outcomes in aging populations.

## Introduction

In the Republic of Korea, the overall number of tuberculosis (TB) cases has been steadily decreasing due to improved public health measures and effective control programs [1]. Nevertheless, the incidence of TB among the elderly population has been increasing, reflecting the rapid aging of the population [2]. Older adults are particularly vulnerable to TB due to age-related immune decline, and they frequently present with atypical clinical manifestations, leading to delays in diagnosis and treatment [3,4]. Additionally, managing TB in elderly patients is particularly challenging owing to the high prevalence of comorbidities, potential drug interactions, and age-related physiological changes, all of which contribute to poorer treatment outcomes [3,4]. Addressing these issues is critical for improving the management and prognosis of TB in this population.

Depression is a common and treatable mental health condition associated with immunosuppression, which is often under-recognized in individuals with TB [5]. Depression may independently increase the risk for TB [6] and adversely impact anti-TB treatment outcomes, including increased mortality [7] and poor treatment adherence [8]. Among elderly individuals, depression is also frequently underdiagnosed, partly due to coexisting physical illnesses and the misconception that

depressive symptoms are a normal part of aging [9]. The dual epidemic of TB and depression in the elderly population of Korea constitutes a serious and emerging public health concern.

Despite its clinical significance, the specific characteristics of depression in elderly individuals with TB have not been sufficiently investigated. Understanding the psychosocial and clinical factors contributing to depression in this vulnerable group is essential for developing targeted interventions and improving TB care outcomes. This study aimed to examine the prevalence of depressive episodes, identify its associated factors, and assess anti-TB treatment outcomes among elderly individuals with TB.

## Methods

### Study design and subjects

We conducted a multicenter prospective cohort study between September 2020 and December 2022 across 13 university-affiliated hospitals in the Republic of Korea. To ensure representativeness, the participating hospitals were distributed across diverse regions of Korea. The cohort included individuals aged ≥ 65 years who were diagnosed with active TB, either microbiologically or clinically. Data on demographic, clinical, and psychosocial characteristics were collected through structured interviews and medical record reviews by trained research personnel.

### Assessing depressive episode

Depressive episodes were assessed using the Patient Health Questionnaire-9 (PHQ-9) at the time of enrollment [10]. The PHQ-9 is a widely used screening tool for evaluating the severity of depression and has been utilized globally since 2002. It consists of nine items that correspond to the diagnostic criteria for major depressive disorder (MDD), measuring the frequency of specific symptoms over the past two weeks. The items assessed anhedonia, depressed mood, sleep disturbance, fatigue, appetite changes, feelings of worthlessness or guilt, concentration difficulties, psychomotor changes, and suicidal thoughts. Each item is rated on a four-point scale: "not at all" (0), "several days" (1), "more than half the days" (2), and "nearly every day" (3), with total scores ranging from 0 to 27, according to the original guidelines to maintain consistency in assessment. We defined a depressive episode using a PHQ-9 cutoff score of ≥10, which has been shown to yield approximately 88% sensitivity and specificity for major depressive disorder [11]. An additional tenth item evaluates functional impairment by asking how difficult these problems have made it difficult for the respondent to perform daily activities at work, at home, or in social relationships. The Korean version of the PHQ-9, which has been validated previously, was used to ensure cultural and linguistic appropriateness [12].

### Data collection

Demographic, socioeconomic, and clinical data were prospectively collected from the enrolled patients using a standardized case report form at the time of study entry. The collected variables included sex, age, body mass index (BMI), history of malignancy, retreatment status (prior TB history), initial TB treatment regimen, extrapulmonary involvement, initial TB-related symptoms, cavitation on chest radiography, sputum acid-fast bacilli smear test results, and rifampicin resistance. BMI was dichotomized at 18.5 kg/m² (<18.5 vs ≥ 18.5) and included as a covariate in the multivariable models. We combined the chest radiographic findings and sputum smear test results to generate a single variable with two strata: mild disease (non-cavitary disease and negative smear results) and severe disease (cavitary disease or positive smear results). Initial TB symptoms were further categorized into three groups: (1) respiratory symptoms such as cough and sputum production; (2) alarming symptoms such as hemoptysis and chest pain; and (3) constitutional symptoms such as fever, weight loss, and night sweats. A predefined checklist was used to systematically collect the initial TB-related symptoms. Prior depression was defined as a clinician-diagnosed depressive disorder under active treatment at enrollment, ascertained by participant report and/or chart documentation.

All participants were followed until the completion of anti-TB treatment. Treatment outcomes were defined based on the Korean TB guidelines, which were aligned with the World Health Organization (WHO) definitions (cured, treatment completed, treatment failed, died, lost to follow-up, not evaluated). Treatment success was defined as the sum of cured and treatment completed. Given the clinical reality that older adults may have delayed response or adverse events necessitating prolonged therapy, we defined treatment success for rifampin-susceptible TB as completion of the prescribed regimen within 9 months. Patients on treatment >9 months were categorized as still-on-treatment and counted among unfavourable outcomes, together with treatment failed, died, lost to follow-up, and not evaluted.

## Statistical analysis

Baseline characteristics are presented for the entire enrolled cohort (n = 361). Continuous variables are expressed as medians with interquartile ranges, while discrete variables are reported as frequencies and percentages. Baseline characteristics were compared between patients with and without depressive episodes, and univariate analyses were conducted using the chi-square test for categorical variables. A univariate analysis using the chi-square test was performed to assess the association between depressive episodes and independent variables. Subsequently, age, sex, and variables with a p-value <0.20 based on a univariable analysis were selected for multivariable binary logistic regression analysis. We performed a sensitivity analysis excluding seven participants with prior-treated depression to evaluate potential bias related to antidepressant therapy status.

Additionally, to identify factors associated with functional impairment, we conducted a separate multivariable logistic regression analysis using responses to Item 10 of the PHQ-9 as the dependent variable. Functional impairment was defined as any level of difficulty (i.e., somewhat, very, or extremely difficult) in performing daily activities due to depressive symptoms. We also conducted a multivariable logistic regression analysis to identify factors associated with suicidal ideation, as measured by Item 9 of the PHQ-9. Suicidal ideation was defined as a non-zero response to Item 9 (i.e., having such thoughts on several days or more during the past two weeks).

Primary treatment-success analyses were restricted to rifampin-susceptible TB. Patients with rifampicin-resistant TB (11/361, 3.0%) were excluded from primary analyses because they receive different, longer regimens; the small number of rifampin-resistant cases would have introduced clinical heterogeneity and confounding by indication. To assess the effect of depressive episodes on treatment success, we conducted univariable and multivariable binary logistic regression analyses, adjusting for variables known to influence TB treatment outcomes. Statistical significance was defined as a p-value of <0.05. All statistical analyses were conducted using the SPSS software (version 17.0; Statistical Product and Services Solutions, Chicago, IL, USA).

## Ethics statement

This study was performed in accordance with the Declaration of Helsinki and approved by the Institutional Review Board of the Catholic University of Korea (IRB No. XC20ENDI0086). Written informed consent was obtained from all participants prior to enrollment.

## Results

Among the 502 participants initially screened for participation in the study, 38 individuals were excluded because they were not diagnosed with active TB, and four individuals withdrew consent, resulting in a cohort of eligible 459 participants with active TB. Of these, 98 individuals did not complete the PHQ-9 and were excluded from the final analysis. The remaining 361 participants comprised the final analytical sample, yielding a response rate of 78.6% (Fig 1).

Table 1 presents the baseline characteristics of the 361 elderly individuals with active TB who were included in the analysis. Participants were 207 male (57.3%) and 154 female (42.7%). The age distribution was as follows: 166 (46.0%)

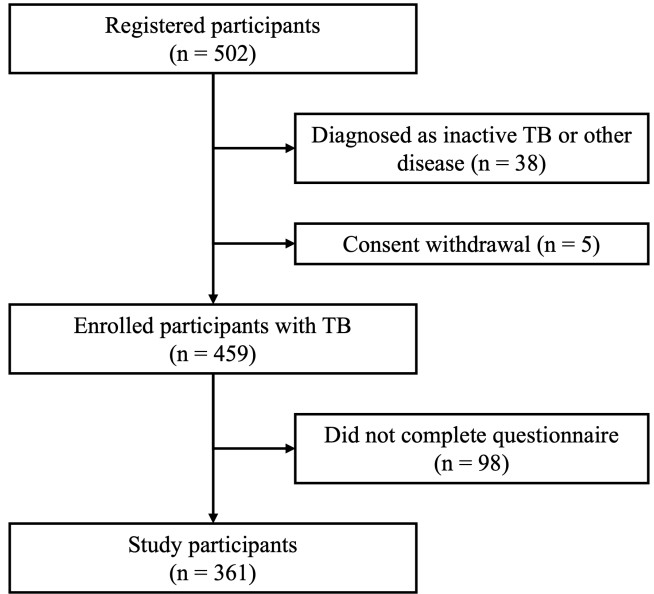

**Fig 1. Flowchart describing the selection of study participants.** Of the 502 registered participants, 38 were excluded due to diagnosis of inactive tuberculosis or other disease, and 5 withdrew consent. A total of 459 participants with tuberculosis were enrolled. Among them, 98 did not complete the questionnaire, resulting in 361 participants included in the final analysis.

were aged 65−74 years, 157 (43.5%) were 75−84 years, and 38 (10.5%) were aged 85 years or older. Only 7 (1.9%) participants reported a prior diagnosis of depression. A total 69 participants (19.1%) were classified as having a depressive episode (PHQ-9 ≥ 10). Compared to those with lower scores, individuals with depressive episodes were significantly more likely to be unemployed (p = 0.006), have higher CCI scores (p = 0.048), and more frequently report initial TB-related symptoms including cough or sputum (p = 0.005), alarming symptoms (p = 0.005), and constitutional symptoms (p < 0.001). The prevalence of depressive episodes by sex was 16.9% in males (35/207) and 22.1% in females (34/154), with no statistically significant difference between sexes (p = 0.217). No significant differences were observed in the age distribution, smoking history, or microbiological/radiological findings between the groups.

In multivariable logistic regression analysis, unemployment was independently associated with higher odds of depressive episodes (adjusted odds ratio [aOR], 5.068; 95% confidence interval [CI], 1.138–22.564) (Table 2). Compared with participants with a CCI score of 0, those with a score of 1–2 had an aOR of 2.819 (95% CI, 1.213–6.551), and those with a score ≥3 had an aOR of 3.195 (95% CI, 1.168–8.739). Participants with cough or sputum had significantly higher odds of depression (aOR, 2.119; 95% CI, 1.147–3.913) than those with constitutional symptoms (aOR, 2.455; 95% CI, 1.395–4.321). Sex was not associated with depressive episodes: relative to males, females had an adjusted odds ratio of 1.168 (95% CI, 0.685–2.076; p = 0.596).

The findings of sensitivity analysis excluding seven participants with prior depression were consistent with the main analysis (S1 Table). Unemployment was strongly associated with depressive episodes (aOR 10.069, 95% CI 1.311–70.303, p = 0.026). Higher comorbidity burden showed a graded association (CCI 1–2 vs 0: aOR 2.720, 95% CI 1.167–6.336, p = 0.020; CCI ≥ 3 vs 0: aOR 3.246, 95% CI 1.182–8.916, p = 0.022). Cough or sputum (aOR 2.008, 95% CI 1.083–3.721, p = 0.027) and constitutional symptoms (aOR 2.356, 95% CI 1.331–4.168, p = 0.003) were also independently associated with higher odds of depressive episodes.

Of 361 participants, 143 (39.6%) had functional impairment. Using PHQ-9 ≥ 10 to define a depressive episode, 69/361 (19.1%) met this threshold and 292 (80.9%) did not (PHQ-9 score of 0–9), comprising 205 with the PHQ-9 score of 0–4

**Table 1. Comparison of baseline characteristics of participants with and without depressive episodes among elderly individuals with tuberculosis.**

| Variables | Depressive episodes | | Total | p-value |
|---|---|---|---|---|
| | No | Yes | | |
| | (n = 292) | (n = 69) | (n = 361) | |
| Sex | | | | |
| Male | 172 (58.9) | 35 (50.7) | 207 (57.3) | 0.217 |
| Female | 120 (41.1) | 34 (49.3) | 154 (42.7) | |
| Age, years | | | | |
| ≤74 | 137 (46.9) | 29 (42.0) | 166 (46.0) | 0.715 |
| 75–84 | 124 (42.5) | 33 (47.8) | 157 (43.5) | |
| ≥85 | 31 (10.6) | 7 (10.1) | 38 (10.5) | |
| Ever smoker [a] | 130 (44.8) | 26 (37.7) | 156 (43.5) | 0.282 |
| Living alone [b] | 74 (25.4) | 20 (29.0) | 94 (26.1) | 0.545 |
| Unemployment [c] | 244 (84.7) | 67 (98.5) | 311 (89.6) | **0.006*** |
| Body mass index [d] | | | | |
| <18.5 kg/m$^2$ | 40 (14.9) | 12 (19.0) | 52 (15.7) | 0.412 |
| ≥18.5 kg/m$^2$ | 229 (85.1) | 51 (81.0) | 280 (84.3) | |
| CCI score | | | | |
| 0 | 74 (25.3) | 8 (11.6) | 82 (22.7) | **0.043*** |
| 1–2 | 171 (58.6) | 47 (68.1) | 218 (60.4) | |
| ≥3 | 47 (16.1) | 14 (20.3) | 61 (16.9) | |
| Depression | 6 (2.1) | 1 (1.4) | 7 (1.9) | 0.743 |
| Diabetes | 87 (29.8) | 24 (34.8) | 111 (30.4) | 0.419 |
| Chronic lung disease | 23 (7.9) | 7 (10.1) | 30 (8.3) | 0.539 |
| Retreatment case | | | | |
| Yes | 51 (17.5) | 12 (17.4) | 63 (17.5) | 0.988 |
| No | 241 (82.5) | 57 (82.6) | 298 (82.5_ | |
| Severe TB disease | 98 (33.6) | 28 (40.6) | 126 (34.9) | 0.271 |
| Rifampicin resistance | 10 (3.4) | 1 (1.4) | 11 (3.0) | 0.391 |
| Initial treatment regimen | | | | |
| HRZE | 268 (91.7) | 66 (95.7) | 334 (92.5) | 0.273 |
| Others | 24 (8.3) | 3 (4.3) | 27 (7.5) | |
| Cough or sputum | 157 (53.8) | 50 (72.5) | 207 (57.3) | **0.005*** |
| Alarming symptoms | 80 (27.4) | 31 (44.9) | 111 (30.7) | **0.005*** |
| Constitutional symptoms | 85 (29.1) | 35 (50.7) | 120 (33.2) | **<0.001*** |

CCI, Charlson Comorbidity Index; TB, tuberculosis; HRZE, isoniazid, rifampin, pyrazinamide, and ethambutol.

Depression was defined using a Patient Health Questionnaire (PHQ-9) threshold of ≥10. Severe TB was defined as the presence of a cavity on chest radiography or a positive sputum smear result. Data are presented as number (%). p-values <0.05 (two-sided) are marked with "*".

Number of missing data: [a] ever smoker (2); [b] Living alone (1); [c] Unemployment (4); [d] Low body mass index (29).

(no depressive episode) and 87 with score 5–9 (mild depressive symptoms). Functional impairment was present in 43/205 (21.0%) in the 0–4 band, 43/87 (49.4%) in the 5–9 band, and 57/69 (82.6%) among those with PHQ-9 ≥ 10 (Fig 2). We conducted a logistic regression analysis to identify the factors associated with functional impairment (S2 and S3 Tables). Although smoking history and constitutional symptoms showed associations in univariable analyses (chi-square tests), only the presence of constitutional symptoms remained significantly associated with functional impairment in the multivariable model (aOR = 2.090, 95% CI: 1.330–3.283).

**Table 2. Multivariable logistic regression analysis of factors associated with depressive episodes among elderly individuals with tuberculosis.**

| | Adjusted OR (95% CI) | P value |
|---|---|---|
| Sex | | |
| Male | Reference | |
| Female | 1.168 (0.685–2.076) | 0.596 |
| Age, years | | |
| ≤74 | Reference | |
| 75–84 | 0.895 (0.486–1.646) | 0.720 |
| ≥85 | 0.846 (0.323–2.216) | 0.734 |
| Unemployment | 5.068 (1.138–22.564) | **0.033*** |
| CCI score | | |
| 0 | Reference | |
| 1–2 | 2.819 (1.213–6.551) | **0.016*** |
| ≥3 | 3.195 (1.168–8.739) | **0.024*** |
| Cough or sputum | 2.119 (1.147–3.913) | **0.016*** |
| Alarming symptoms | 1.747 (0.989–3.088) | 0.055 |
| Constitutional symptoms | 2.455 (1.395–4.321) | **<0.001*** |

OR, odds ratio; CI, confidence interval; CCI, Charlson Comorbidity Index.

A depressive episode was defined as a Patient Health Questionnaire (PHQ-9) score of ≥10. Variables included in the model were selected based on their clinical relevance and the results of the univariate analysis. p-values <0.05 (two-sided) are marked with "*".

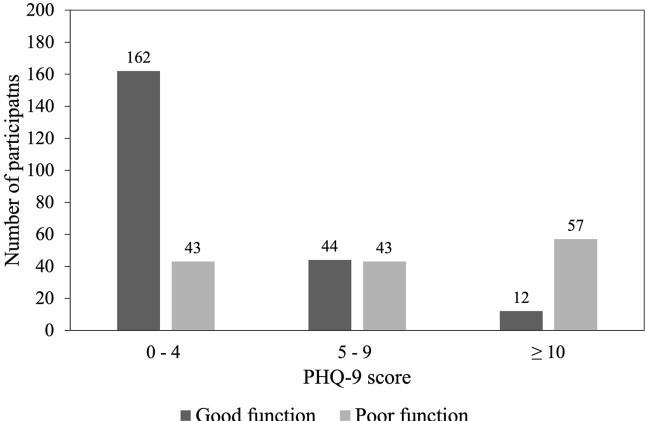

**Fig 2. Bar chart showing the distribution of functional status (good vs. poor function) by Patient Health Questionnaire-9 (PHQ-9) score categories among study participants.** In the PHQ-9 score group 0–4, 162 participants had good function and 43 had poor function. In the 5–9 group, 44 had good function and 43 had poor function. In the ≥10 group, 12 had good function and 57 had poor function. The proportion of participants with poor function increases as PHQ-9 scores increase.

Suicidal ideation was reported by 71/361 participants (19.7%) participants in the total study participants (S4 Table). In the multivariate logistic regression analysis, the presence of alarming symptoms was significantly associated with suicidal ideation (aOR = 2.239, 95% CI: 1.285–3.903) (S5 Table). Although prior depression diagnosis and higher CCI scores (≥3) showed borderline associations with suicidal ideation, they did not reach statistical significance.

Of 361 eligible participants, 350 with rifampicin-susceptible TB were included in the primary treatment-success analyses, after excluding 11 with rifampicin-resistant TB. The overall treatment success rate for rifampin-susceptible TB was 76.3% (267/350) (Tables 3 and S6). Compared to those without depressive episodes, individuals with depressive episodes had significantly lower treatment success rates (79.1% [223/282] vs. 64.7% [44/68], p = 0.012). In the univariable logistic regression analysis, participants with depressive episodes had significantly lower odds of treatment success (odds ratio [OR], 0.449; 95% confidence interval [CI]: 0.249–0.861). This association remained significant after adjusting for sex and age in Model 2 (aOR, 0.449; 95% CI, 0.249–0.811) and after further adjusting for prior TB treatment history, diabetes, AFB smear positivity, and cavitary lesions in Model 3 (aOR, 0.478; 95% CI, 0.261–0.878).

## Discussion

In this cohort of elderly individuals with TB, depressive episodes were common and associated with multiple clinical and functional outcomes. Depressive episodes were independently associated with unemployment, a higher comorbidity burden, and the presence of coughing and constitutional TB-related symptoms. Functional impairment increases with depression severity. Importantly, participants with depressive episodes had significantly lower odds of treatment success, even after adjusting for clinical covariates. These findings emphasize the importance of assessing mental health in this population and highlight the need to identify clinical risk factors that contribute to the development of depression.

The prevalence of depression among patients with TB varies considerably depending on the region and population. A systematic review and meta-analysis reported a pooled prevalence of 38.4% when assessed using the PHQ-9, with even higher rates observed in patients with multidrug-resistant TB and in females [13]. Unlike previous studies, our study focused exclusively on elderly individuals with TB, allowing us to identify distinct associated factors. Notably, unemployment was significantly associated with depression, suggesting that remaining socially engaged through employment or daily activities may serve as a buffer against depressive symptoms among older adults. Furthermore, initial TB-related

**Table 3. Association between depressive symptoms and treatment success among elderly individuals with rifampicin-susceptible tuberculosis.**

| Variables | Total | Treatment success | P value | Model #1 | Model #2 | Model #3 |
|---|---|---|---|---|---|---|
| | n (col%) | n (row%) | | OR (95% CI) | Adjusted OR (95% CI) | Adjusted OR (95% CI) |
| Total | 350 (100%) | 267 (76.3%) | | | | |
| Depressive episode | | | | | | |
| No | 282 (80.6) | 223 (79.1) | 0.012 | Reference | Reference | Reference |
| Yes | 68 (19.4) | 44 (64.7) | | **0.485 (0.273 - 0.861) *** | **0.449 (0.249 - 0.811) *** | **0.478 (0.261 - 0.878) *** |
| Sex | | | | | | |
| Male | 155 (44.3) | 117 (75.5) | 0.055 | | Reference | Reference |
| Female | 195 (55.7) | 150 (76.9) | | | **1.993 (1.158 - 3.431) *** | **1.863 (1.067 - 3.252) *** |
| Age, years | | | | | | |
| ≤74 | 160 (45.7) | 129 (80.6) | 0.042 | | Reference | Reference |
| 75–84 | 154 (44.0) | 116 (75.3) | | | 0.647 (0.371 - 1.131) | 0.620 (0.349 - 1.104) |
| ≥85 | 36 (10.3) | 22 (61.1) | | | **0.324 (0.146 - 0.723) *** | **0.333 (0.145 - 0.763) *** |
| Diabetes | 109 (31.1) | 76 (69.7) | 0.052 | | | 0.592 (0.344 - 1.019) |
| Prior TB treatment | 60 (17.1) | 39 (65.0) | 0.024 | | | **0.511 (0.270 - 0.966) *** |
| Severe TB disease | 121 (34.6) | 80 (66.1) | 0.001 | | | **0.474 (0.280 - 0.801) *** |

TB, tuberculosis.

Values are expressed as odds ratio and 95% confidence interval. Model 1 was the result of the univariate logistic regression analysis. Models 2 and 3 show the results of the multivariate logistic regression analysis. p-values <0.05 (two-sided) are marked with "*".

symptoms, particularly cough, sputum production, and constitutional symptoms, were more common among those with depression, possibly because of heightened illness perception and psychological distress. These findings underscore the importance of patient education and social support in fostering a sense of control over the disease and in mitigating its emotional burden.

The ninth item of the PHQ-9 is designed to screen for suicidal ideation. In the context of South Korea's rapidly aging population, suicide among older adults is a growing public health concern. Psychological distress following TB diagnosis may serve as a potent trigger for suicidal thoughts or behaviors. Individuals with TB have a higher prevalence of suicidality compared to the general population [14]. Suicide accounted for approximately 2.2% to 8.4% of all TB-related deaths, with the pooled prevalence of suicidal ideation being 8.5% and suicide attempts 3.1% among TB patients. In our study, nearly one in five participants reported suicidal ideation. Among the various clinical and sociodemographic variables, the presence of alarming TB symptoms, such as dyspnea, hemoptysis, or chest pain, was the only factor significantly associated with suicidal ideation in the multivariable analysis. The WHO's "LIVE LIFE" suicide prevention strategy emphasizes the importance of early identification, assessment, and management of individuals exhibiting suicidal behavior [15]. In line with this guidance, systematic depression screening and timely referral to mental health professionals for patients with suicidal thoughts should be integrated into routine TB care.

Functional impairment is a core diagnostic criterion for major depressive disorder, which requires depressive symptoms to cause significant disruption in social, occupational, or daily functioning. Even when depressive symptoms are mild, clinical intervention may be warranted if functional impairment is present. In our study, the proportion of participants with poor functioning increased with depression severity. Notably, even among those without depression or with only mild symptoms, 40% reported functional impairments. This may reflect the baseline vulnerability of our study population, which consisted entirely of older adults who may have already experienced limitations in their daily activities. Interestingly, constitutional symptoms were the only factor significantly associated with functional impairment in our study, suggesting that early TB-related symptoms, such as fatigue, fever, or weight loss, may substantially affect patients' ability to function, particularly in elderly individuals. Given that functional impairment may persist even in cases of mild depression, appropriate clinical interventions should be considered to improve the prognosis of elderly individuals with TB.

In the present study, depressive episodes were significantly associated with lower TB treatment success rates. Mental disorders, including depression, may negatively affect TB treatment outcomes by increasing the risk of loss to follow-up and non-adherence [16]. These effects may be more pronounced in older adults, such as those in our study population, who are likely to face multiple comorbidities, increased vulnerability to adverse drug reactions, and reduced motivation to adhere to lengthy treatment regimens. To improve TB treatment outcomes in such populations, a patient-centered approach is essential. Integrating mental healthcare with TB services is critical. Healthcare workers should actively assess the psychosocial needs of people with TB, recognize coexisting mental health conditions, and support treatment for both TB and depression in an integrated manner. This dual-focus strategy may enhance adherence, reduce premature treatment discontinuation, and improve clinical outcomes.

This study has several limitations. First, depression was assessed using self-reported questionnaires rather than clinical psychiatric evaluations, which may have introduced a response bias, potentially leading to either underestimation or overestimation of depressive symptoms. Second, we lacked standardized data on antidepressant medication type, dose, duration, and pre-TB depression severity. Active treatment may alleviate symptoms and bias associations toward the null, whereas treatment indication may reflect greater baseline severity and bias estimates away from the null. Future studies should capture psychotropic therapy longitudinally. Third, as the study focused exclusively on elderly patients with TB, the findings may not be generalizable to younger populations or those with different comorbid conditions. Further studies involving more diverse cohorts are required to validate our results. Fourth, we lacked participant-level data on region (urban vs rural), adverse effects, and additional/adjunct therapies, which may result in residual confounding. Prospective, standardized capture of these variables would enable proper adjustment and tests of effect modification.

## Conclusion

Depressive symptoms are common among elderly patients with TB. Routine mental health assessment and early intervention should be considered as a part of comprehensive TB care in this population. Proactive mental health management may improve adherence, enhance the quality of life, and potentially lead to better outcomes.

## Supporting information

**S1 Table. Multivariable predictors of depressive episodes (PHQ-9 ≥ 10) after excluding participants with prior depression.**
(DOCX)

**S2 Table. Baseline characteristics of participants stratified by functional status.**
(DOCX)

**S3 Table. Multivariable logistic regression analysis to assess factors associated with functional impairment.**
(DOCX)

**S4 Table. Baseline characteristics of participants stratified by suicidal ideation.**
(DOCX)

**S5 Table. Multivariable logistic regression analysis to assess factors associated with suicidal ideation.**
(DOCX)

**S6 Table. Distribution of treatment outcomes by depressive episode (PHQ-9 ≥ 10) among 350 participants with rifampin-susceptible tuberculosis.**
(DOCX)

## Author contributions

**Conceptualization:** Hyung Woo Kim, Ju Sang Kim, Yun-Jeong Jeong, Yoolwon Jeong, Jinsoo Min.

**Data curation:** Jin Woo Kim, Yong Hyun Kim, Heayon Lee, Yeonhee Park, Jee Youn Oh, Hyeon-Kyoung Koo, Ock-Hwa Kim, Yong-Soo Kwon, Won Yeon Lee, Jinsoo Min.

**Formal analysis:** Ahran Kim, Yoolwon Jeong, Jinsoo Min.

**Funding acquisition:** Jinsoo Min.

**Investigation:** Hyung Woo Kim, Ju Sang Kim, Jin Woo Kim, Yong Hyun Kim, Heayon Lee, Yeonhee Park, Jee Youn Oh, Hyeon-Kyoung Koo, Ock-Hwa Kim, Yun-Jeong Jeong, Yong-Soo Kwon, Won Yeon Lee, Jinsoo Min.

**Methodology:** Hyung Woo Kim, Ju Sang Kim, Yoolwon Jeong, Jinsoo Min.

**Project administration:** Hyung Woo Kim, Ju Sang Kim, Jinsoo Min.

**Resources:** Jinsoo Min.

**Software:** Jinsoo Min.

**Supervision:** Jinsoo Min.

**Validation:** Yoolwon Jeong, Jinsoo Min.

**Visualization:** Ahran Kim, Jinsoo Min.

**Writing – original draft:** Ahran Kim, Jinsoo Min.

**Writing – review & editing:** Ahran Kim, Hyung Woo Kim, Ju Sang Kim, Jin Woo Kim, Yong Hyun Kim, Heayon Lee, Yeonhee Park, Jee Youn Oh, Hyeon-Kyoung Koo, Ock-Hwa Kim, Yun-Jeong Jeong, Yong-Soo Kwon, Won Yeon Lee, Yoolwon Jeong, Jinsoo Min.

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
