## [Decision Letter · Decision Letter 0]

18 Sep 2025

Dear Dr. Min,

Several aspects require clarification and refinement before the manuscript can be considered further. These include ensuring consistency in reported sample sizes and denominators, clarifying methodological choices such as exclusion criteria and outcome definitions, and providing more explicit justification for the handling of prior depression and treatment-resistant TB cases. Additional explanatory detail is also needed where results appear counterintuitive, such as the lower frequency of depressive episodes among those with a history of depression.

From an analytical standpoint, the study would benefit from either incorporating additional relevant variables (e.g., body size, regional context, treatment type, regimen, resistance patterns, and management of comorbidities) or, if this is not feasible, acknowledging the limitations clearly. Explicit reporting of sex-based comparisons is also encouraged to improve interpretability.

Finally, we ask that you enhance clarity of presentation by resolving inconsistencies across the text, figures, and tables, marking statistically significant results more clearly, declaring GenAI use, and making minor improvements to the language and style.

We look forward to receiving your revised manuscript.

Kind regards,

Muhammad Shahzad Aslam, Ph.D.,M.Phil., Pharm-D

Academic Editor

PLOS ONE

2. In the online submission form, you indicated that [The ownership of the primary datasets lies with the Korea National Institute of Health (KNIH). The de-identified datasets generated and/or analysed during the current study can be made available for replication purposes upon reasonable request. Interested researchers should first contact the corresponding author, who will provide detailed instructions for submitting a request to the KNIH, including required documents such as a brief study proposal, institutional review board (IRB) approval, and a data use agreement. Access will be granted only with permission from the KNIH and will be limited to the scope of the approved research.].

Additional Editor Comments:

Please clarify whether the reported 19.7% prevalence was observed in the total population or restricted to the subgroup with depressive episodes.

Ensure that the cited references directly support the statement on poor adherence; replace or revise as needed.

Please explain why WHO’s culture-based definition of “cure” was not applied. If culture data are available, provide the breakdown of patients meeting “cure” and report on treatment failure.

justify clearly Exclusion of rifampicin-resistant TB patients

Clarify whether subjects with prior depression were receiving antidepressant therapy, and discuss the potential impact on your findings.

Address inconsistencies in patient counts. Please reconcile and correct throughout.

Consider whether body size (BMI/weight), region (urban vs rural), adverse effects, treatment type (initial vs. retreatment), adjunct therapies, TB regimen type, and resistance patterns can be incorporated into the analysis. If not feasible, explain why.

Mark statistically significant p-values in all tables (univariate and multivariate analyses) for ease of reading.

Reviewers' comments:

Reviewer's Responses to Questions

**Comments to the Author**

1. Is the manuscript technically sound, and do the data support the conclusions?

Reviewer #1: Yes

Reviewer #2: Yes

2. Has the statistical analysis been performed appropriately and rigorously?

Reviewer #1: Yes

Reviewer #2: Yes

3. Have the authors made all data underlying the findings in their manuscript fully available?

Reviewer #1: Yes

Reviewer #2: Yes

4. Is the manuscript presented in an intelligible fashion and written in standard English?

Reviewer #1: Yes

Reviewer #2: Yes

Reviewer #1: Overall Summary:

This is a well-designed and scientifically sound cohort study showing that depressive episodes negatively affect treatment outcomes in elderly tuberculosis patients. The study is important for aging populations and should be accepted after minor revision.

Manuscript Number: PONE-D-25-44015

Title: Depressive episode and treatment outcomes in elderly individuals with tuberculosis: a prospective cohort study in Korea

Journal: PLOS ONE

1. Originality and Appropriateness

The manuscript presents results of an original prospective multicenter cohort study conducted in Korea (2020–2022). The findings have not been previously published elsewhere, and the study clearly addresses an important public health problem by examining the relationship between depressive episodes and tuberculosis treatment outcomes in elderly patients. The topic fits the scope of PLOS ONE. This research is particularly timely and needed for aging populations. After World War II, many individuals suffered from poor health due to lack of medical care in their youth. Now, as they reach older age, these long-term health consequences exacerbate disease burden. Addressing mental health in this context is of high importance.

2. Methodological Rigor

Study design: Prospective multicenter cohort across 13 hospitals, representative of different regions in Korea.

Sample: 361 elderly participants (≥65 years) with active TB. Clear inclusion/exclusion process shown in the flowchart.

Assessment: Depression was measured with the validated Korean version of the PHQ-9, with a cutoff ≥10 defining depressive episode.

Analysis: Logistic regression was applied to identify risk factors and associations with treatment success. Adjusted models included important covariates (age, sex, comorbidities, TB severity).

Ethics: IRB approval obtained (C19ONDI0458, XC20ENDI0086). Written informed consent documented.

The methodology is adequate, technically sound, and well described. Analyses are appropriate and reproducible. However, one suggestion for improvement: sex should be analyzed and presented more fully, not only by using “female” as a reference category. Explicit comparisons between males and females would make interpretation clearer for readers.

3. Results

Prevalence: 19.1% of participants had a depressive episode.

Risk factors: Unemployment (aOR 5.07), higher Charlson comorbidity score, cough, and constitutional symptoms.

Functional impairment: Increased with depression severity.

Suicidal ideation: Reported by 19.7%; associated with alarming TB symptoms.

Treatment outcomes: Depressive episode significantly reduced treatment success (64.7% vs. 79.1%, p=0.012). Independent association confirmed (aOR 0.478, 95% CI 0.261–0.878).

The results are clearly presented with supportive tables and figures. The statistical evidence supports the conclusions.

4. Discussion and Conclusions

The discussion appropriately interprets findings:

- Depression is common among elderly TB patients and linked with poor adherence, functional decline, and suicidality.

- Highlights the importance of routine depression screening at TB diagnosis.

- Discusses public health relevance and implications for integrated care.

- Limitations are acknowledged: self-reported PHQ-9, limited generalizability beyond elderly cohort.

Conclusions are supported by data, not overstated, and aligned with the study’s scope. The authors should consider revising the reporting of sex differences to make male–female comparisons explicit rather than leaving readers to infer.

5. Writing and Presentation

The article is written in standard English, intelligible and well structured.

Abstract accurately reflects the study.

Tables and figures are appropriate and clear.

References are up to date and relevant.

Minor language polishing could further enhance readability, but the manuscript already meets PLOS ONE standards.

6. Ethics and Data Availability

Ethics approval and informed consent are properly documented.

Data availability is described: de-identified datasets are accessible upon request via KNIH with IRB approval and data use agreement. This is compliant with PLOS ONE data policy.

No competing interests declared.

Funding disclosure complete.

7. Recommendation

The manuscript is scientifically sound, ethically compliant, and meets PLOS ONE criteria.

I recommend acceptance after minor revision, specifically encouraging the authors to (1) polish English slightly, and (2) report sex differences explicitly (male vs. female) for clearer interpretation.

Reviewer #2: Kim and colleagues have performed a logistic regression analysis to identify significant factors associated with depressive episodes and TB treatment outcome. Notably, the authors assessed functional impairment and suicidal ideation using different scales, which strengthens the analysis by highlighting factors that distinctly impact each. The findings that unemployment, higher comorbidity burden, and constitutional symptoms were associated with depressive episodes, and that alarming symptoms were linked to suicidal ideation, are important observations. Furthermore, the association of depressive episodes with poor treatment success adds clinical relevance.

The work is valuable; however, I have identified several areas for improvement that would enhance the clarity, robustness, and interpretability of the study:

Major Comments

1. Line 54–55: Please clarify whether suicidal ideation (19.7%) was observed in the total population or specifically in the subgroup with depressive episodes.

2. Line 77: References 7 and 8 do not appear to contain data directly related to poor treatment adherence. Please verify and provide appropriate references.

3. Line 127–128: According to WHO, a negative sputum culture defines cure or treatment success. Why was this not applied? If available, please provide the breakdown of patients who met the “cure” definition. Including data on treatment failure would also be highly informative.

4. Line 148: Please provide the rationale for excluding rifampicin-resistant TB patients.

5. Line 173: Including subjects with prior depression may bias the analysis. Were any of these subjects receiving antidepressant therapy? Please clarify.

6. Results: Consider incorporating additional variables into the analysis to strengthen robustness, such as:

Body size (weight, BMI)

Region (urban vs. rural)

Adverse effects or pre-treatment adverse effects

Treatment type (initial vs. retreatment)

Additional therapies (e.g., immunotherapy, management of depression)

TB regimen type

Resistance pattern (XDR, MDR, etc.)

7. Table 1: It is counterintuitive that patients with prior depression have fewer depressive episodes. Could this be due to antidepressant therapy or other management strategies? Please comment.

8. Table 1: The methods indicate rifampicin-resistant TB patients were excluded, yet this variable appears in the table. Please explain.

9. Line 203–205: In Table 2, 292 patients are listed under “no depressive episodes,” but here the number is 205. Please clarify this discrepancy.

10. Line 225: The total sample size is 361, but treatment success is reported for 350. Was treatment outcome assessed only in this subset? Please clarify why 11 subjects were excluded.

Minor Comments

* Please mark statistically significant p-values in both univariate and multivariate analyses with an asterisk for improved readability.

**Do you want your identity to be public for this peer review?** For information about this choice, including consent withdrawal, please see our Privacy Policy

Reviewer #1: **Yes: ** Dr. Jing Liang

Reviewer #2: No

---

## [Author Response · Author response to Decision Letter 1]

6 Oct 2025

Editor Comments:

[Comment #1] Please clarify whether the reported 19.7% prevalence was observed in the total population or restricted to the subgroup with depressive episodes.

[Respone #1]

We thank the editor for this helpful suggestion. We have clarified that the 19.7% prevalence refers to suicidal ideation among all 361 participants assessed with PHQ-9 item 9. We have made the following changes.

“Suicidal ideation was observed in 19.7% (71/361) of all participants …” [Page 3, line 54 – 55]

“Suicidal ideation was reported by 71/361 participants (19.7%) participants in the total study participants.” [Page 11, line 220 – 221]

Similarly, to clarify that the proportion of functional impairment was based on 143 of the total 361 participants, we revised the text as follows.

“Functional impairment was reported in 39.6% (143/361) of all participants …” [Page 3, line 53 – 54]

[Comment #2] Ensure that the cited references directly support the statement on poor adherence; replace or revise as needed.

[Response #2]

Thank you for the helpful comment. We reviewed the citations supporting the statement on poor adherence and made the following changes. We replaced the previous Reference 8 with a publication that explicitly evaluates non-adherence/loss to follow-up among TB patients with depressive symptoms.

[8] Panati D, Chittooru CS, Madarapu YR, Gorantla AK: Effect of depression on treatment adherence among elderly tuberculosis patients: A prospective interventional study. Clinical Epidemiology and Global Health 2023, 22: 101338.

We updated the sentence and in-text citation at line 77 accordingly.

“… adversely impact anti-TB treatment outcomes, including increased mortality [7] and poor treatment adherence [8].” [Page 4, line 75 – 76]

[Comment #3] Please explain why WHO’s culture-based definition of “cure” was not applied. If culture data are available, provide the breakdown of patients meeting “cure” and report on treatment failure.

[Response #3]

Thank you for this important point. In Korea, treatment outcome categories follow WHO guidance. Consistent with those definitions, we defined treatment success as the sum of “cured” (bacteriologically confirmed pulmonary TB with negative smear/culture at treatment completion and on ≥1 prior occasion) plus “treatment completed” (completed treatment without evidence of failure but lacking the required end-of-treatment/prior negative results).

Because many older adults in routine care have incomplete serial culture data—owing to smear-negative/paucibacillary disease, extrapulmonary involvement, or missed end-of-treatment specimens—we prespecified treatment success for drug-susceptible TB as completion of the prescribed regimen within 9 months, aligned with national practice for the elderly.

Patients who remained on therapy beyond 9 months were categorized as still-on-treatment and counted under unfavourable outcomes, together with treatment failed, loss to follow-up, death, and not evaluated.

“ … (cured, treatment completed, treatment failed, died, lost to follow-up, not evaluated). Treatment success was defined as the sum of cured and treatment completed. Given the clinical reality that older adults may have delayed response or adverse events necessitating prolonged therapy, we defined treatment success for rifampin-susceptible TB as completion of the prescribed regimen within 9 months. Patients on treatment >9 months were categorized as still-on-treatment and counted among unfavourable outcomes, together with treatment failed, died, lost to follow-up, and not evaluted.” [Page 6, line 129 – 135]

In response to the reviewer’s request, we now report the full distribution of treatment outcomes in the analytic rifampin-susceptible TB population (n=350), stratified by depressive episode (PHQ-9 ≥10), in the supplmentary.

Supplemental table 6. Distribution of treatment outcomes by depressive episode (PHQ-9 ≥10) among 350 participants with rifampin-susceptible tuberculosis.

Treatment outcome Depressive episodes Total

No Yes

(n = 282) (n = 68) (n = 350)

n % n % n %

Treatment success 223 79.1 44 64.7 267 76.3

Treatment failed 4 1.4 2 2.9 6 1.7

Lost to follow-up 6 2.1 2 2.9 8 2.3

Died 15 5.3 6 8.8 21 6.0

Not evaluated 16 5.7 7 10.3 23 6.6

Still-on-treatment 18 6.4 7 10.3 25 7.1

[Comment #4] justify clearly Exclusion of rifampicin-resistant TB patients

[Response #4]

Thank you for the comment. In our cohort, rifampicin-resistant TB (MDR/RR-TB) comprised 11 of 361 patients (3.0%). RR-TB patients receive fundamentally different, longer regimens and adherence-support strategies, introducing substantial clinical heterogeneity relative to drug-susceptible TB. Given the very small RR-TB sample size and the non-comparability of treatment pathways, including these patients in the primary analysis would yield unstable estimates and potential confounding by indication. We therefore excluded RR-TB from the primary treatment-success models.

We updated and added sentences in the Method and Results sections as follows.

“Primary treatment-success analyses were restricted to rifampin-susceptible TB. Patients with rifampicin-resistant TB (11/361, 3.0%) were excluded from primary analyses because they receive different, longer regimens; the small number of rifampin-resistant cases would have introduced clinical heterogeneity and confounding by indication.” [Page 7, line 154 – 157]

“Of 361 eligible participants, 350 with rifampicin-susceptible TB were included in the primary treatment-success analyses, after excluding 11 with rifampicin-resistant TB.” [Page 13, line 247 – 248]

[Comment #5] Clarify whether subjects with prior depression were receiving antidepressant therapy, and discuss the potential impact on your findings.

[Response #5]

Thank you for this important point. In our dataset, “prior depression” was operationalized as a clinician-diagnosed depressive disorder under active treatment at enrollment (ascertained by self-report and/or chart documentation).

We did not systematically record antidepressant type, dose, duration, or pre-TB depression severity. Because treatment can reduce symptoms, the observed associations may be biased toward no effect. Conversely, the fact that some patients were treated may indicate more severe baseline depression, which could bias estimates away from no effect. Prospective studies should track psychotropic therapy over time.

We added following sentences in the Method and Limitation sections.

“Prior depression was defined as a clinician-diagnosed depressive disorder under active treatment at enrollment, ascertained by participant report and/or chart documentation.” [Page 6, line 124 – 126]

“Second, we lacked standardized data on antidepressant medication type, dose, duration, and pre-TB depression severity. Active treatment may alleviate symptoms and bias associations toward the null, whereas treatment indication may reflect greater baseline severity and bias estimates away from the null. Future studies should capture psychotropic therapy longitudinally.” [Page 18, line 318 – 322]

To assess potential bias, we conducted a sensitivity analysis excluding seven participants with prior-treated depression.

“We performed a sensitivity analysis excluding seven participants with prior-treated depression to evaluate potential bias related to antidepressant therapy status.” [Page 7, line 145 – 146]

The results were materially unchanged compared with the primary analysis, which were presented in the supplementary and the Result section as follows.

Supplemental table 1. Multivariable predictors of depressive episodes (PHQ-9 ≥10) after excluding participants with prior depression

Adjusted OR (95% CI) P value

Sex

Male Reference

Female 1.190 (0.668 – 2.121) 0.554

Age, years

74 Reference

75 – 84 0.883 (0.477 – 1.634) 0.691

85 0.821 (0.313 – 2.151) 0.688

Unemployment 10.069 (1.311 – 70.303) 0.026

CCI score

0 Reference

1 – 2 2.720 (1.167 – 6.336) 0.020*

3 3.246 (1.182 – 8.916) 0.022*

Cough or sputum 2.008 (1.083 – 3.721) 0.027*

Alarming symptoms 1.747 (0.982 – 3.082) 0.058

Constitutional symptoms 2.356 (1.331 – 4.168) 0.003*

The summary of the exclusion sensitivity analysis was presented in the Result section as follows.

“The findings of sensitivity analysis excluding seven participants with prior depression were consistent with the main analysis (S1 Table). Unemployment was strongly associated with depressive episodes (aOR 10.069, 95% CI 1.311–70.303, p=0.026). Higher comorbidity burden showed a graded association (CCI 1–2 vs 0: aOR 2.720, 95% CI 1.167–6.336, p=0.020; CCI ≥3 vs 0: aOR 3.246, 95% CI 1.182–8.916, p=0.022). Cough or sputum (aOR 2.008, 95% CI 1.083–3.721, p=0.027) and constitutional symptoms (aOR 2.356, 95% CI 1.331–4.168, p=0.003) were also independently associated with higher odds of depressive episodes.” [Page 12, line 219 – 225]

[Comment #6] Address inconsistencies in patient counts. Please reconcile and correct throughout.

[Response #6]

Thank you for the careful reading. After a line-by-line check, we confirm that no numerical errors were present; however, we recognize that our wording could prompt confusion. We have revised the text, as detailed below.

“Using PHQ-9 ≥10 to define a depressive episode, 69/361 (19.1%) met this threshold and 292 (80.9%) did not (PHQ-9 score of 0–9), comprising 205 with the PHQ-9 score of 0–4 (no depressive episode) and 87 with score 5–9 (mild depressive symptoms). Functional impairment was present in 43/205 (21.0%) in the 0–4 band, 43/87 (49.4%) in the 5–9 band, and 57/69 (82.6%) among those with PHQ-9 ≥10 (Figure 2).” [Page 13, line 226 – 230]

“Of 361 eligible participants, 350 with rifampicin-susceptible TB were included in the primary treatment-success analyses, after excluding 11 with rifampicin-resistant TB.” [Page 13, line 247 – 248]

[Comment #7] Consider whether body size (BMI/weight), region (urban vs rural), adverse effects, treatment type (initial vs. retreatment), adjunct therapies, TB regimen type, and resistance patterns can be incorporated into the analysis. If not feasible, explain why.

[Response #7]

Thank you for the suggestion. We implemented the following additions/clarifications.

(1) BMI: We created a new variable using the 18.5 kg/m² cutoff (<18.5 vs ≥18.5) and included it in the models. BMI was not statistically significant and did not alter the main estimates.

(2) Regimen type: We added a binary indicator for receipt of the standard HRZE regimen at initiation (HRZE vs non-HRZE). This variable was not statistically significant and did not change conclusions.

(3) Resistance pattern: The number of rifampicin-resistant cases was very small (n=11), precluding meaningful subdivision (e.g., XDR-TB).

(4) Treatment type: We already captured this via prior TB history; for clarity we renamed the variable to “retreatment case” and retained it in the multivariable models.

The contents of the “data collection”subsection were updated as follows.

“The collected variables included sex, age, body mass index (BMI), history of malignancy, retreatment status (prior TB history), initial TB treatment regimen, extrapulmonary involvement, initial TB-related symptoms, cavitation on chest radiography, sputum acid-fast bacilli smear test results, and rifampicin resistance. BMI was dichotomized at 18.5 kg/m² (<18.5 vs ≥18.5) and included as a covariate in the multivariable models.” [Page 6, line 114 – 118]

Across these robustness checks, results were materially unchanged relative to the main analysis.

We did not collect participant-level region (urban vs rural), adverse effects (including pre-treatment AEs), or additional therapies (e.g., immunotherapy, active depression management/psychotherapy). These variables were therefore not included in the models.

“Fourth, we lacked participant-level data on region (urban vs rural), adverse effects, and additional/adjunct therapies, which may result in residual confounding. Prospective, standardized capture of these variables would enable proper adjustment and tests of effect modification.” [Page 18, line 325 – 327]

[Comment # 8] Mark statistically significant p-values in all tables (univariate and multivariate analyses) for ease of reading.

[Response #8]

Thank you for the suggestion. We have updated all statistical tables to flag statistically significant p-values with an asterisk for clarity.

Reviewer #1:

Overall Summary:

This is a well-designed and scientifically sound cohort study showing that depressive episodes negatively affect treatment outcomes in elderly tuberculosis patients. The study is important for aging populations and should be accepted after minor revision.

Manuscript Number: PONE-D-25-44015

Title: Depressive episode and treatment outcomes in elderly individuals with tuberculosis: a prospective cohort study in Korea

Journal: PLOS ONE

1. Originality and Appropriateness

The manuscript presents results of an original prospective multicenter cohort study conducted in Korea (2020–2022). The findings have not been previously published elsewhere, and the study clearly addresses an important public health problem by examining the relationship between depressive episodes and tuberculosis treatment outcomes in elderly patients. The topic fits the scope of PLOS ONE. This research is particularly timely and needed for aging populations. After World War II, many individuals suffered from poor health due to lack of medical care in their youth. Now, as they reach older age, these long-term health consequences exacerbate disease burden. Addressing mental health in this context is of high importance.

2. Methodological Rigor

Study design: Prospective multicenter cohort across 13 hospitals, representative of different regions in Korea.

Sample: 361 elderly participants (≥65 years) with active TB. Clear inclusion/exclusion process shown in the flowchart.

Assessment: Depression was measured with the validated Korean version of the PHQ-9, with a cutoff ≥10 defining depressive episode.

Analysis: Logistic regression was applied to identify risk factors and associations with treatment success. Adjusted models included important covariates (age, sex, comorbidities, TB severity).

Ethics: IRB approval obtained (C19ONDI0458, XC20ENDI0086). Written informed consent documented.

The methodology is adequate, technically sound, and well described. Analyses are appropriate and reproducible. However, one suggestion for improvement: sex should be analyzed and presented more fully, not only by using “female” as a reference category. Explicit comparisons between males and females would make interpretation clearer for readers.

3. Results

Prevalence: 19.1% of participants had a depressive episode.

Risk factors: Unemployment (aOR 5.07), higher Charlson comorbidity score, cough, and constitutional symptoms.

Functional impairment: Increased with depression severity.

Suicidal ideation: Reported by 19.7%; associated with alarming TB symptoms.

Treatment outcomes: Depressive episode significantly reduced treatment success (64.7% vs. 79.1%, p=0.012). Independent association confirmed (aOR 0.478, 95% CI 0.261–0.878).

The results are clearly presented with supportive tables and figures. The statistical evidence supports the conclusions.

4. Discussion and Conclusions

The discussion appropriately interprets findings:

- Depression is common among elderly TB patients and linked with poor adherence, functional decline, and suicidality.

- Highlights the importance of routine depression screening at TB diagnosis.

- Discusses public health relevance and implications for integrated care.

- Limitations are acknowledged: self-reported PHQ-9, limited generalizability beyond elderly cohort.

Conclusions are supported by data, not overstated, and aligned with the study’s scope. The authors should consider revising the reporting of sex differences to make male–female comparisons explicit rather than leaving readers to infer.

5. Writing and Presentation

The article is written in standard English, intelligible and well structured.

Abstract accurately reflects the study.

Tables and

---

## [Decision Letter · Decision Letter 1]

20 Oct 2025

Depressive episode and treatment outcomes in elderly individuals with tuberculosis: a prospective cohort study in Korea

PONE-D-25-44015R1

Dear Dr. Min,

We’re pleased to inform you that your manuscript has been judged scientifically suitable for publication and will be formally accepted for publication once it meets all outstanding technical requirements.

Kind regards,

Muhammad Shahzad Aslam, Ph.D.,M.Phil., Pharm-D

Academic Editor

PLOS ONE

Additional Editor Comments (optional):

Reviewers' comments:

Reviewer's Responses to Questions

**Comments to the Author**

Reviewer #1: All comments have been addressed

Reviewer #2: All comments have been addressed

2. Is the manuscript technically sound, and do the data support the conclusions?

Reviewer #1: Yes

Reviewer #2: Yes

3. Has the statistical analysis been performed appropriately and rigorously?

Reviewer #1: Yes

Reviewer #2: Yes

4. Have the authors made all data underlying the findings in their manuscript fully available?

Reviewer #1: Yes

Reviewer #2: Yes

5. Is the manuscript presented in an intelligible fashion and written in standard English?

Reviewer #1: Yes

Reviewer #2: Yes

Reviewer #1: I have carefully reviewed the revised manuscript and the authors’ detailed responses. The revision has fully addressed all prior reviewer and editor comments. The manuscript now meets the scientific, ethical, and editorial standards required by PLOS ONE. The study is methodologically sound, the presentation is clear, and the revisions improve transparency and rigor. I recommend acceptance of the manuscript.

Editor’s Comments

1.Clarification of suicidal ideation prevalence (19.7%)

→ Revised and clarified that this percentage refers to all 361 participants. Clear statements added to Abstract and Results sections.

2.Reference accuracy on treatment adherence

→ Reference #8 updated to Panati et al., Clin Epidemiol Glob Health (2023), directly supporting poor adherence in TB patients with depression.

3.WHO definition of “cure” and reporting treatment failure

→ The revised manuscript now explicitly follows WHO guidance, clarifies the rationale for using a 9-month cutoff for treatment success, and provides a detailed outcome table (Supplemental Table 6).

4.Exclusion of rifampicin-resistant TB patients

→ Justified based on clinical heterogeneity and small sample size (3.0%, n=11). Explanation added to both Methods and Results sections.

5.Prior depression and antidepressant therapy

→ Defined as clinician-diagnosed and under active treatment; discussion of bias and sensitivity analysis excluding 7 such participants were added. Results unchanged.

6.Consistency of patient counts

→ Clarified that PHQ-9 <10 group (n=292) includes both 0–4 and 5–9 subgroups; explicit breakdown provided.

7.Additional variables suggested (BMI, regimen, region, etc.)

→ BMI and regimen type were added to models; other variables (region, AE data, adjunct therapy) were unavailable and acknowledged as a limitation.

8.Marking significant p-values

→ All tables now include asterisks (*) for significant results for improved readability.

All editorial concerns have been fully resolved.

Reviewer #1 Comments

1.Male–female comparison clarity

→ Added explicit sex-based comparisons in text and tables, including male/female counts and adjusted odds ratio (female a OR = 1.168, 95% CI 0.685–2.076).

2.Language quality

→ English refined for clarity, formatting standardized, and terminology harmonized (e.g., rifampicin).

Reviewer #1’s suggestions have been thoroughly addressed.

Reviewer #2 Comments

All ten major comments and one minor comment have been carefully addressed:

1. Clarified suicidal ideation base (19.7%) → Corrected and clarified.

2. Updated references for poor adherence → Completed.

3. WHO cure definition and treatment outcome table → Fully revised and justified.

4.Exclusion of RR-TB → Clear rationale added.

5.Prior depression and antidepressant therapy → Defined, analyzed, and discussed with sensitivity analysis.

6.Added BMI and other covariates → Included and explained, unchanged conclusions.

7.Prior depression fewer episodes issue → Explained due to active treatment and small sample size.

8.RR-TB in Table 1 explanation → Clarified as descriptive baselines for all 361 participants.

9.Count discrepancy (292 vs 205) → Corrected with explicit PHQ-9 breakdown.

10.Treatment success denominator (350) → Explained exclusion of 11 RR-TB participants.

11.Mark significant p-values → All tables revised accordingly.

Each issue raised by Reviewer #2 is now completely resolved.

Reviewer #2: (No Response)

**Do you want your identity to be public for this peer review?** For information about this choice, including consent withdrawal, please see our Privacy Policy

Reviewer #1: **Yes: ** Dr. Jing Liang

Reviewer #2: No

---

## [Editor Report · Acceptance letter]

PONE-D-25-44015R1

PLOS ONE

Dear Dr. Min,

I'm pleased to inform you that your manuscript has been deemed suitable for publication in PLOS ONE. Congratulations! Your manuscript is now being handed over to our production team.

Kind regards,

on behalf of

Dr. Muhammad Shahzad Aslam

Academic Editor

PLOS ONE